# Very-large-scale reconfigurable intelligent surfaces for dynamic control of terahertz and millimeter waves

Yury Malevich [1,2], M. Said Ergoktas [1,2,3], Gokhan Bakan[1,2], Pietro Steiner[1,2] & Coskun Kocabas [1,2,4] ✉

Unlocking the potential of terahertz (THz) and millimeter (mm) waves for next generation communications and imaging applications requires reconfigurable intelligent surfaces (RIS) with programmable elements that can manipulate the waves in real-time. Realization of this technology has been hindered by the lack of efficient THz electro-optical materials and scalable THz semiconductor platform. Here, by merging graphene-based THz modulators and the thin-film transistor (TFT) technology, we demonstrate very-large-scale (>300000 pixels) spatial light modulator with individually addressable subwavelength pixels. We demonstrate electronically programmable reflection and transmission patterns of THz light over a large area with unprecedent levels of uniformity and reproducibility. To highlight the potential of these devices, we demonstrate a single pixel mm-wave camera capable of imaging metallic objects. Furthermore, we demonstrate dynamic beam steering with reconfigurable direction pattern. We anticipate that these results will provide realistic pathways to structure THz waves for applications in non-invasive THz imaging and next generation THz communications.

Due to material and technological challenges, the terahertz (THz) (0.3–3 THz) and sub-THz (90–300 GHz) region remains a relatively underutilized portion of the electromagnetic spectrum, despite its wide range of promising applications in communication[1–4], sensing[5,6], and imaging technologies[6–10]. The advancement of future THz technologies necessitates devices capable of generating spatio-temporal patterns of THz light[11]. Studies on THz optoelectronics have led to several successful demonstrations using active metamaterials[12–15], liquid crystals[16–18], phase-change materials[19–21], plasmonic modulators[22], and photoinduced electrons[23]. Although these results are impressive at the single-device level, they encounter challenges in large-scale system-level integration, critical for unlocking new possibilities for scalable THz technologies[15]. A promising alternative involves controlling high-mobility charges on graphene[24–29] via electrostatic gating in a transistor configuration. These devices leverage

the tuneable Drude-like metallic response of graphene at THz frequencies. However, the developmental stage of these devices is still nascent compared to those operating in the visible spectrum. Overcoming these developmental hurdles would bridge existing technological gaps for emerging THz technologies.

In this work, we introduce an integration scheme that incorporates large arrays of graphene-based THz modulators, allowing programmable control over reflection and transmission across the wide THz and mm-wave spectrum. Figure 1a depicts the structure of a pixel, highlighting the key functional layers: graphene, electrolyte, and backplane electronics. In this architecture, the THz active layer consists of bilayer graphene, produced by chemical vapor deposition on copper foils, and subsequently transferred onto a thin polymer sheet (~70 μm PET, see Supporting Materials Fig. S1). A 5 μm-thick, porous membrane infused with an ionic liquid electrolyte is laminated between the

[1]Department of Materials, University of Manchester, Manchester, UK. [2]National Graphene Institute, University of Manchester, Manchester, UK. [3]Department of Physics, University of Bath, Claverton Down, Bath, UK. [4]Henry Royce Institute for Advanced Materials, University of Manchester, Manchester, UK. ✉e-mail: coskun.kocabas@manchester.ac.uk

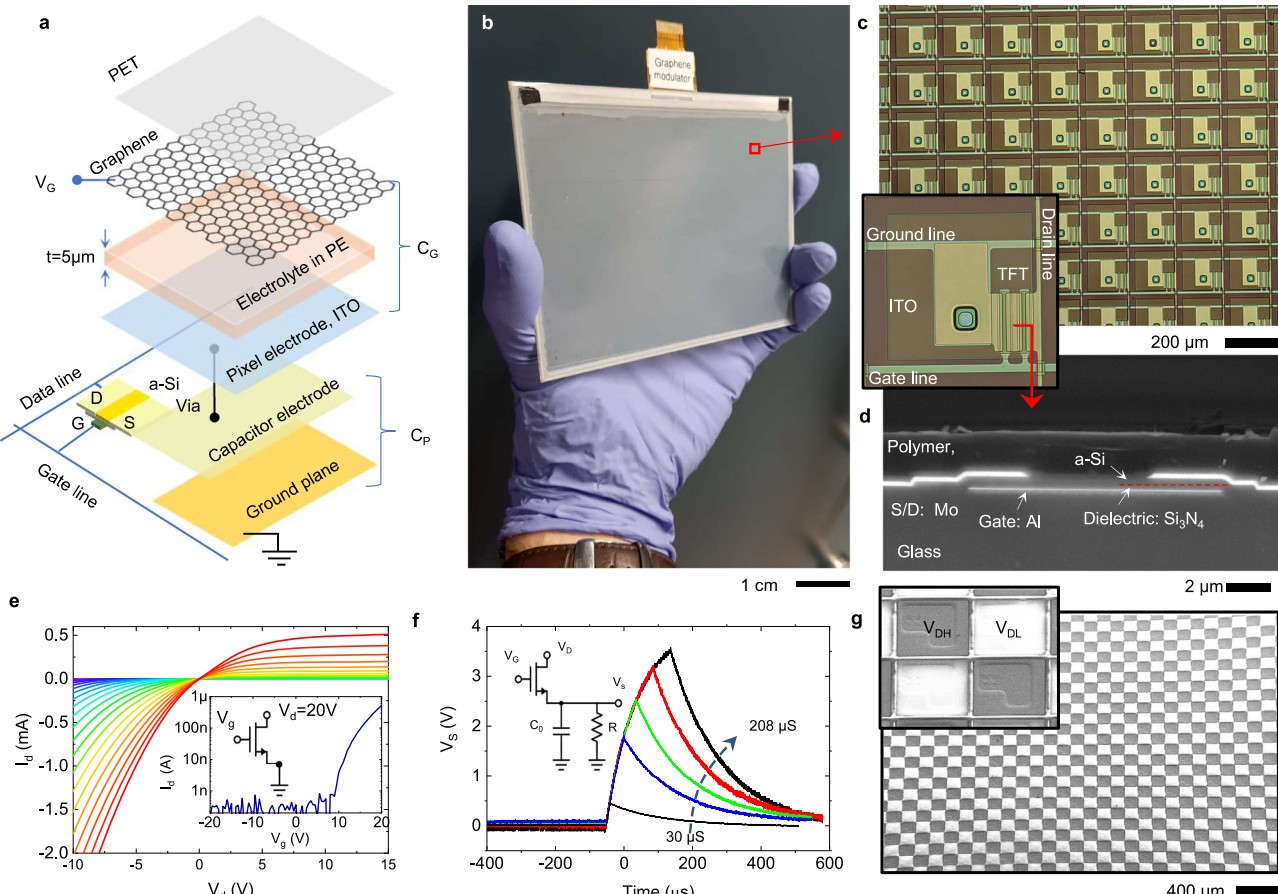

**Fig. 1 | Reconfigurable THz surface. a** Schematic of the pixel structure, comprising laminated layers including a graphene top electrode, an electrolyte layer, and a back pixel electrode. Common voltage, $V_G$, is applied to the graphene layer to compensate unintentional doping, thereby enabling the device to operate near the Dirac point. The charge on the capacitor is regulated by the duration of gate voltage pulses applied to the gate line. **b** Photograph of the fabricated device consisting of an active-matrix array of 640 × 480 pixels. A binary voltage pattern ($V_{DH}$, $V_{DL}$) is produced by a chip-on-glass display driver controlled by an external micro-controller. **c** Photograph of the TFT back panel, displaying the sub-wavelength pixels, each consisting of a top ITO electrode connected to the pixel capacitor with a via through the polymer layer. The inset illustrates the unit cell with a size of

185 μm. A double back-gated a-Si thin-film transistor, with a channel length of 5 μm and a width of 75 μm, governs the charge/voltage on the pixel capacitor, which is formed between the middle capacitor electrode and the ground plane. **d** Scanning electron micrograph revealing the cross-section of the a-Si thin-film transistor. **e** Representative output ($I_d$ vs $V_d$) and transfer ($I_d$ vs $V_g$) characteristics of the n-type a-Si TFT. **f** Analysis of the pixel voltage's dependence on the width of the applied gate pulses. The inset presents the circuit model of the pixel, incorporating the probe resistance (1 MΩ) and capacitance (16 pF), resulting in a 1.5 ms charging time. **g** Scanning electron microscope image of the surface, with charging contrast induced by the alternating voltages of $V_{DH}$ and $V_{DL}$ on the pixels.

graphene layer and the partially transparent indium tin oxide (ITO) pixel electrode. In this device, the thickness of the electrolyte layer is critical to minimize electrostatic crosstalk between pixels (Supporting Materials Fig. S2) enabling area selective charge accumulation on a continuous graphene layer. The charge accumulated on the pixel capacitor (formed between the middle electrode and ground plane, as shown in Fig. 1a) dictates the local hole/electron density on the graphene layer through electrolyte gating, thereby modulating THz reflection and transmission. We applied an additional negative bias voltage ($V_G$) on the graphene layer to offset the shift of the Dirac point due to unintentional doping during the fabrication process.

## Results

### Integration of graphene devices with TFT arrays

The back-plane electronics, illustrated in Fig. 1b, c, utilize active-matrix technology to regulate the local voltage of individually addressable sub-wavelength pixels within a dense array of 640 rows and 480 columns covering 12 × 9 cm² area. The inset in Fig. 1c displays an enlarged image of a pixel, which contains an *n*-type amorphous-Si TFT with a double-back-gate configuration (Fig. 1d). The TFT, pixel capacitor, and voltage lines are covered by a 2 μm thick polymer layer. The top ITO

layer maintains direct contact with the electrolyte and is connected to the pixel electrode through a metalized via. The scalability of our approach is primarily constrained by the capabilities of modern display technology, which enables the production of very large-scale, dense arrays of TFTs on both rigid and flexible substrates (See Supporting Materials Fig. S3). In the manuscript, we only characterized the 0.3 megapixel device. However, our method is capable of scaling the device size to multiple megapixels with suitable backplane electronics.

Figure 1e presents the output and transfer characteristics of the *n*-type a-Si TFT. The charge on the pixel capacitor is controlled by applying positive voltage pulses on the gate while switching between high ($V_{DH}$) and low drain ($V_{DL}$) voltage with scan rate of 200 Hz. Due to the current saturation at positive drain voltage, the charging and discharging currents display a significant difference, which is compensated by the duration of applied gate pulses. The drain and gate voltage are scanned over the modulator array by a chip-on-glass display driver capable of generating a binary voltage pattern for $V_{DH}$ and $V_{DL}$. The effective pixel voltage is then determined by the width of the gate pulse (varied from 30 to 208 μs), as depicted in Fig. 1f. To visualize the generated voltage pattern at single pixel level, we recorded an electron microscope image of the operating TFT array using a

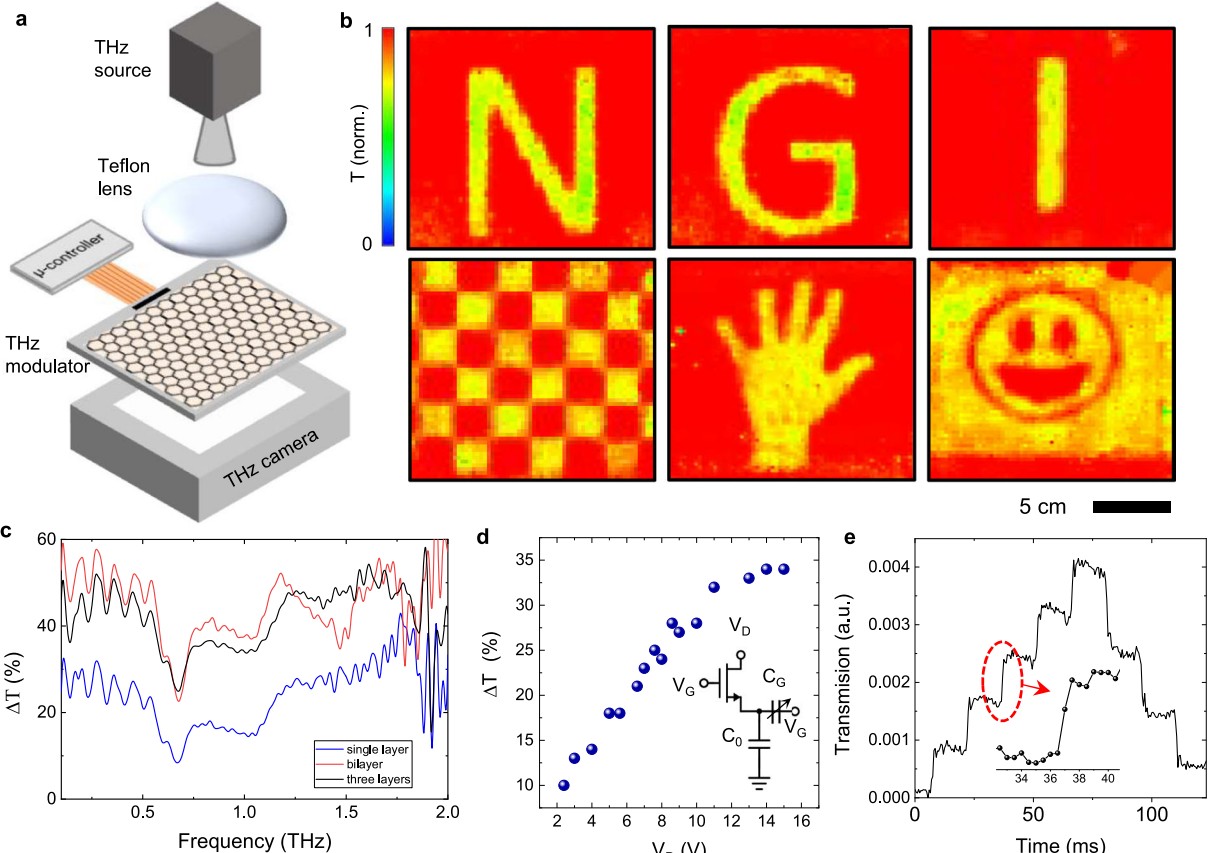

**Fig. 2 | Programmable THz transmission patterns. a** Schematic of the experimental setup used for measuring transmission pattern using a 64 × 64 pixels THz camera. **b** Representative images generated by the THz modulator array. **c** Spectrum of the THz modulation for 3 different devices with single, double, and three layers of graphene **d** Drain voltage dependence of the modulation. The inset shows the circuit model where $C_O$ and $C_G$ represent the pixel capacitance and the

voltage-dependent capacitance of the electrical double layer, respectively. Note that due to the dynamic charging of the pixel capacitor, the drain voltage does not directly appear on the pixel. **e** Variation of the transmission at 100 GHz after applying consecutive charging and discharging pulses. The inset shows the transition region with switching time around 1 ms.

secondary electron detector. The contrast in Fig. 1g results from the voltage difference between the pixels. Pixels with negative voltage generate a greater number of secondary electrons, thereby enhancing the contrast significantly.

Our reconfigurable surfaces operate in both transmission and reflection modes, offering dynamic control over THz and millimeter waves. For the transmission demonstrations, we utilized a 100 GHz source as our camera is specifically optimized for this frequency. For the reflection measurements, we employed a time-domain system capable of performing spectroscopic measurements across a wide range, extending up to 3 THz. First, we analyzed the transmission mode by measuring the programmable spatio-temporal patterns of THz transmission using a sensor array (Terasense, 64 × 64 array with 1.5 mm pixel pitch) and illumination with a collimated 100 GHz beam generated by an IMPATT diode (Fig. 2a). Figure 2b shows representative images of letters N, G, I (the initials of the National Graphene Institute) and different transmission patterns (see Supporting Materials Fig. S4). The spatial resolution of these recorded images is limited by pixel size of the sensor array and the wavelength of the source ($\lambda$ ~ 3 mm). It is important to note that THz transmission of the modulator is around 30% at 100 GHz (see Supporting Materials Fig. S5). The modulation values presented in Fig. 2 are normalized by the transmitted intensity.

To optimize THz transmission modulation, we fabricated modulators with single, double, and three layers of graphene. Figure 2c shows a comparison of the THz modulation spectrum derived from

various devices using a time-domain THz spectrometer. The device with the double layer of graphene marked a significant enhancement over the one with a single layer; however, the introduction of a third layer resulted in considerable insertion loss without any improvement in modulation efficiency. This diminishing performance is presumably attributable to the inefficient gating of the subsequent graphene layers. All devices presented in this work were fabricated using double-layer graphene.

The drain voltage and the duration of the gate pulse are the two main control parameters of the modulator array. Figure 2d illustrates the dependence of the THz modulation (@1 THz) on the drain voltage. To quantify the charging time of the pixel, we applied pulses of $V_{DH}$ and $V_{DL}$ and monitored the transmission with a fast zero-bias Schottky diode attached to a horn antenna (Fig. 2e). We observed that the charging and discharging time can be reduced to 1 ms with short gate pulses of 30 µs (see the inset in Fig. 2e and Supporting materials Figs. S6–S8). To further understand the limitations of the device, we performed electromagnetic simulations of the device using modified Poisson–Nernst–Plank (PNP) equations to include polarization, diffusion, and steric effects of the ionic liquid electrolyte (See Supporting Materials Fig. S9)[30–32]. We have assessed the time response of charging of pixel capacitors and contribution of steric effects and clustering dynamics to this charging process. In the numerical simulations, we transiently applied voltage to one pixel electrode and grounded the graphene layer while monitoring the charge accumulation. These results suggest that the steric effects in IL electrolyte determine the

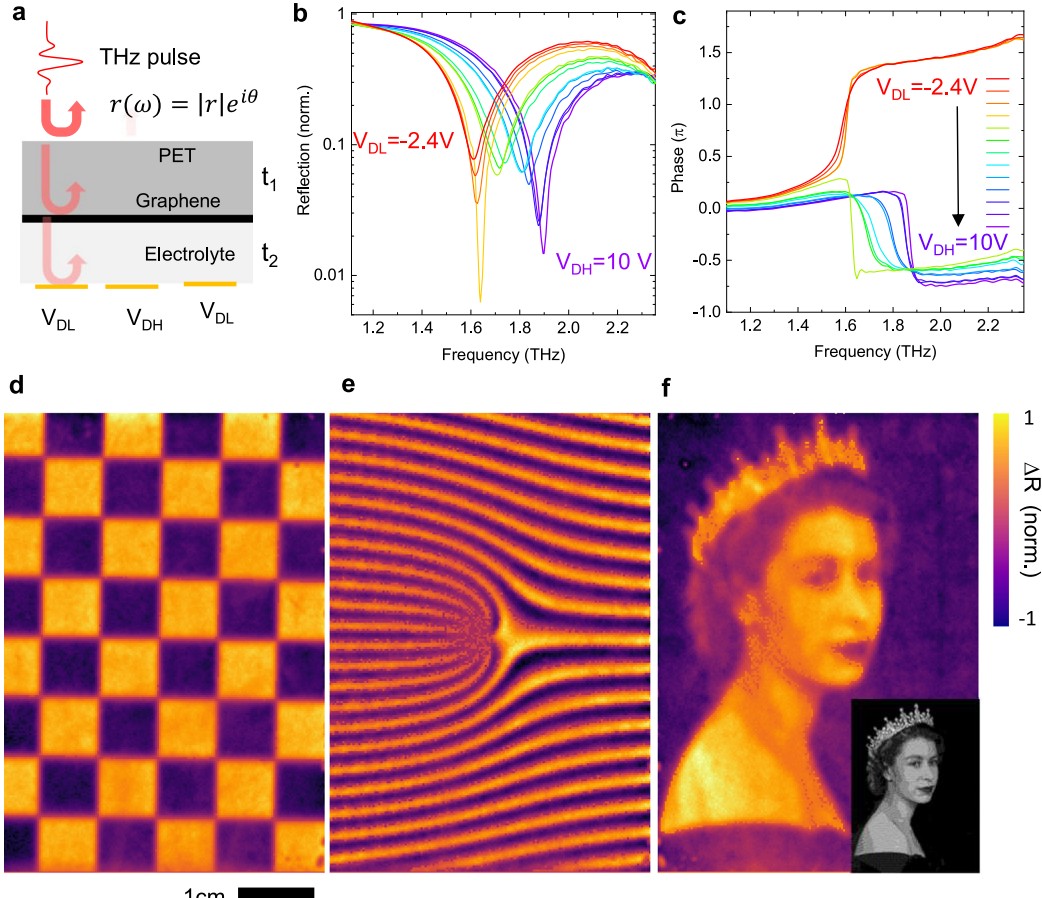

**Fig. 3 | Reconfigurable THz reflective surfaces. a** Structure of the modulator showing the interfaces responsible for the multiple THz reflection resulting a tuneable resonance bavior. **b, c** shows the variation of the intensity and phase spectrum as the pixel voltages switched between $V_{DL} = -2.4V - V_{DH} = 10$ V and recorded with equal time intervals. The step-like phase modulation is due to a topological phase transition enabled by the reflection singularity. **d–f** show THz reflectivity map of a checkerboard, a grayscale fork diffraction grating and a grayscale portrait of Queen Elizabeth II. The inset shows the digitized binary image using 3 × 3 supercell. (By permission of © Royal Collection Enterprises Limited 2025 | Royal Collection Trust).

charging time of the pixel capacitor. By engineering the size of the ions, the switching speed of the modulator can be further improved. However, the complexity of the modulator introduces several technical factors that limit the update rate of the patterns on the device. The primary constraint is the communication capacity of the chip driver, which is restricted by the speed of the serial communication interface. Additionally, factors such as the row scan rate and electronic delays within the circuit further impact the device's maximum achievable speed. Employing more advanced display drivers could significantly reduce the update time, bringing it closer to the intrinsic limit set by electrostatic gating (see Supporting Materials Fig. S10).

**Reconfigurable THz reflection**

Next, we investigate the reconfigurable THz reflectivity. To obtain large phase and intensity modulation, we have designed the device to operate around a reflection singularity, which emerges from the interference of multiple reflections from the top polymer layer ($t_1 = 70\,\mu$m), graphene and TFT array[33]. In this configuration, graphene behaves as a tuneable reflectivity mirror controlling the ratio of the interfering components (Fig. 3a). This multilayer structure behaves as a tuneable coupled cavity. To observe the resonance around at THz frequencies, we increased the thickness of the electrolyte layer to $t_2 = 25\,\mu$m. This configuration enables continuous tunability of the resonance frequency between $f_{max} \sim \frac{1}{t_1}$ and $f_{min} \sim \frac{1}{t_1 + t_2}$. When graphene is reflective (@$V_{DH}$), the resonance frequency is determined by the top polymer coating ($t_1 \sim 70\,\mu$m), however around the Dirac point

(@$V_{DL}$), the graphene layer is transparent therefore the resonance is mainly defined by the total thickness ($t_1 + t_2 \sim 95\,\mu$m). We mounted the device on a motorized xy-stage and measured the time-domain THz reflection spectrum using a 4-mirror reflection head. Figure 3b, c shows the variation of reflection amplitude and phase spectrum as the pixel voltage changes from $V_{DL} = -2.4$ V to $V_{DH} = 10$ V. During this process, the resonance frequency was observed to vary from 1.6 to 1.9 THz. In addition to the continuous tunability, the reflection spectrum goes through a perfect absorption indicating a topological switching clearly evidenced by a step-like jump in the phase spectrum (Fig. 3c). Besides the tuneable phase shift, this phase jump provides a modulation up to $2\pi$. It is important to note that this phase jump is not a branch error in phase measurements, but a geometrical phase due to winding a singularity point. To characterize reconfigurable THz patterns, we generated binary images on the modulator array and performed a raster scan of the THz reflectivity to map these patterns. The spatial resolution of our THz spectrometer is approximately 600 $\mu$m, which is sufficient to resolve fine details in the THz reflectivity patterns. Figure 3d shows a reflectivity map ($\Delta R = R(V_{DH}) - R(V_{DL})$) at 1.5 THz of checkerboard pattern generated by a binary image on the modulator. It is also possible to generate a grayscale THz pattern by defining a supercell by grouping a 3 × 3 array of pixels with binary voltage (see Supporting Materials Fig. S11). In this configuration, a 10-step grayscale value can be achieved by averaging the binary values of the pixels in the supercell. Figure 3e shows a reflection map of a fork diffraction grating obtained by superimposing linear diffraction

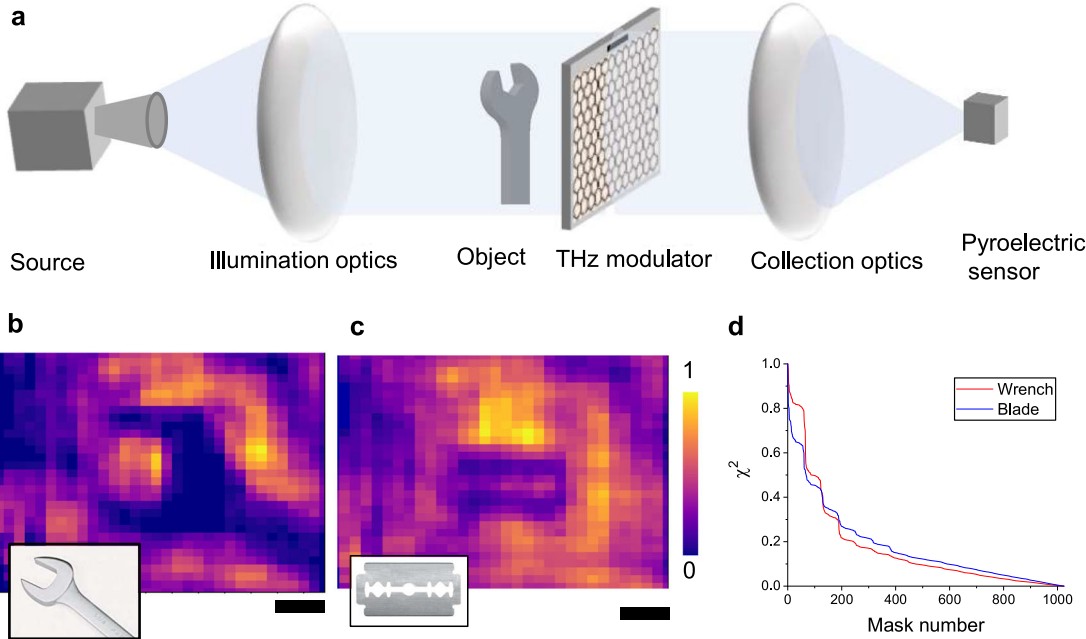

**Fig. 4 | Single-pixel millimeter wave camera. a** Experimental setup used for the single pixel camera consisting of an mm-wave source (100 GHz), illumination and collection optics, reconfigurable modulator array and a single pyroelectric sensor. The metallic object is placed in front of the modulator. **b, c** show the reconstructed images of a wrench and razor blade. The scale bars are 2 cm. **d** shows the convergence of the error function as a function of mask number used in the reconstruction algorithm.

grating pattern with a spiral phase structure. Besides its intricate structure, this pattern contains a dislocation with a topological charge, enabling diffracted beams with vortices, which can be used in various applications, like beam shaping and holography. To test the limits of our approach we also mapped a complex high-resolution pattern, a portrait of Queen Elizabeth II (Fig. 3f) showing the grayscale THz reflectivity.

## Single pixel THz imaging

To showcase a practical application of these large-scale modulators, we built a single-pixel camera capable of imaging concealed metallic objects. Figure 4a shows the experimental setup used for the system. The modulator array is illuminated by a collimated beam of 100 GHz light. The total transmitted light is collected and measured with a pyroelectric sensor. A metallic object placed in a paper envelope is positioned in front of the modulator. The core working principle of the camera is based on compressed sensing algorithm[13,34–36]. This algorithm enables the reconstruction of the spatial information of the object ($X$, $1 \times 1024 = 32 \times 32$ the vector representing the transmittance of the object) by measuring a series of structured transmission patterns generated by the modulator. Here, the shadow of the object on the modulator is reconstructed by changing the transition pattern of the modulator.

We created a set of Hadamard transmission masks ($M$, $1024 \times 1024$ matrix containing the mask set) on the modulator and recorded the transmitted intensity ($I$, $1 \times 1024$ intensity array). Hadamard matrices are particularly well-suited for imaging applications due to their orthogonal rows, which yields minimal cross-talk between patterns. Using a Hadamard matrix of size 1024, we generated 1024 patterns of $32 \times 32$ images. Each mask encodes specific spatial information about the object, and the transmitted intensity was recorded for each pattern. The recorded intensity values were then used to reconstruct the object ($X$) through a computational decoding process that leverages the orthogonality of the Hadamard matrix. For each mask, we measured the differential modulation by changing the pixel voltage between $V_{DL}$ and $V_{DH}$ (see Supporting Materials Fig. S12). This

approach generated positive or negative values, which is required to construct the orthogonal set of Hadamard masks to decode patterns as $X = M^{-1} I$. Figure 4b, c shows reconstructed $32 \times 32$ pixel images of a wrench and a razor blade. It is important to note that the resolution of the imaging system is limited by the wavelength of the source ($\lambda = 3$ mm) used for the illumination. Notably, the cutout width of razor blade, ~2 mm, is visible in the reconstructed image. The graph in Fig. 4d shows the convergence of the image error function ($\chi^2$) with the mask number used in the reconstruction process. It should be noted that the modulation performance of our devices in reflection mode is higher than that in transmission mode. However, due to the current limitations of our time-domain THz system, we were unable to conduct single-pixel imaging in reflection mode. Our current setup primarily supports transmission mode, restricting our ability to fully explore and demonstrate the imaging potential of the reflection mode. Addressing these technical challenges in future studies would enable more detailed spectroscopic image images of objects.

## Dynamic beam steering

Our devices enable dynamic beam steering and beam forming, showcasing possible uses for THz communication. To demonstrate this capability, we generated a dynamic diffraction pattern on the modulator and imaged the resulting diffracted beams. Figure 5 illustrates the experimental setup used for dynamic beam steering through the device using a binary diffraction pattern. In the experimental setup, a 6-in. Teflon lens with a focal length of 15 cm was used to focus the 100 GHz diffracted beams onto a THz camera. By changing the period of the diffraction patterns, we were able to control the direction of the diffracted beam. For instance, applying binary grating patterns with periods ranging from 8 mm to 20 mm resulted in distinct diffracted beams with tuneable angles, as shown in the images.

Additionally, our device has the capability to generate complex THz beams with orbital angular momentum (OAM). When a fork diffraction pattern with a dislocation singularity was applied to the modulator (see Supporting materials Fig. S13), an input Gaussian beam was transformed into donut-shaped Laguerre–Gaussian beams with

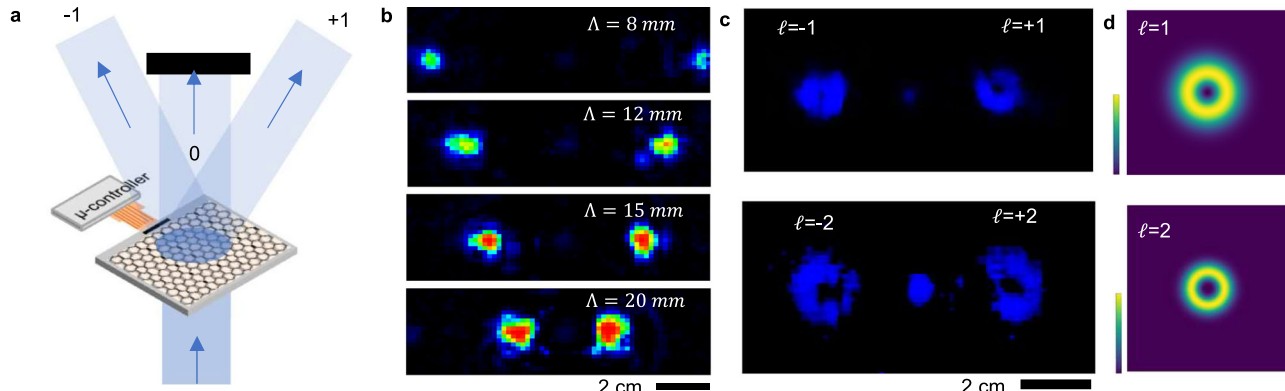

**Fig. 5 | Dynamic beam steering via diffraction. a** Experimental configuration used for the imaging of the diffracted beams from the device. The diffracted beams are focused on the camera using 6" Teflon lens with $f = 15$ cm. **b** Images of the diffracted beams generated by the modulator with binary grating pattern with period changing from 8 to 20 mm. **c** Images of the diffracted beams with orbital angular momentum of $l = 1$ and $l = 2$. The mask generated on modulator converts a Gaussian beam to a donut-shaped Laguerre–Gaussian beams with a phase vortex. **d** Calculated intensity profile of a Laguerre–Gaussian beams with $l = 1$ and $l = 2$.

phase vortices corresponding to OAM values of $l = \pm l$. The characteristic donut shape of the diffracted beam serves as evidence of the vortex and associated angular momentum. By varying the order of the dislocation (i.e., the charge of the singularity) in the fork diffraction pattern, we controlled the angular momentum of the diffracted beam, with Fig. 5c showcasing a diffraction pattern corresponding to $l = \pm 2$. For comparison, we calculated the intensity profiles of Laguerre–Gaussian beams, as shown in Fig. 5d. For comparison, we also calculated the intensity profiles of Laguerre–Gaussian beams, which are shown in Fig. 5d. As the angular momentum of the beam increases, the size of the vortex becomes larger, reflecting the expanding vortex structure. This demonstration highlights the ability of the device to dynamically manipulate beam profiles, enabling more complex optical operations.

As conclusion, by merging graphene modulators and the active-matrix TFT technology, we demonstrate large-area reconfigurable surfaces with sub-wavelength programmable elements that can manipulate THz- and mm-waves in real time. These devices can create reconfigurable spatio-temporal patterns of THz light both in transmission and reflection modes. We demonstrate that the reflection performance of these devices can be significantly improved by operating around a reflection singularity which creates large phase modulation. It should be noted that the observed phase modulation generated with these devices are associated with large insertion losses due to the tuneable THz absorption of the graphene layer. Our results suggest that these surfaces can be used as a spatial light modulator to dynamically steer THz light or to create complex beams, which is a critical capability required for next-generation communication systems. Furthermore, the proof-of-concept demonstration of the single-pixel THz imaging and the dynamic beam steering via diffraction highlight potential for applications in security, medical imaging, and industrial inspection, where non-invasive inspection methods are essential.

## Methods
### Fabrication of the modulator
We purchased A4-size single-layer graphene, synthesized via CVD on copper foil, from MCK Tech Co. Ltd. A 70 μm-thick Polyethylene terephthalate (PET) polymer layer was laminated onto the copper foil at 130 °C. Following this step, the copper layer was etched away in a 0.1 M ammonium persulfate solution, and the remaining graphene on PET was rinsed with deionized water. The modulator array was created by laminating a 25 or 5 μm-thick porous polyethylene layer containing ionic liquid electrolyte (DEME TFSI) onto the TFT backplane. Electrical contacts were deposited at the corners of the graphene layer and connected to the voltage source on the modulator. The custom design

TFT backplane electronic is obtained from a third-party display manufacturer. The backplane electronics are fabricated on a 500 μm thick glass substrate using a conventional a-Si TFT process. The design consists of four electrode layers (back electrode, gate electrode, source-drain electrode, and pixel electrode), two dielectric layers (capacitor dielectric and gate dielectric), and one passivation layer. The first back electrode layer is formed by sputtering and patterning a 100 nm thick aluminum layer. A 250 nm thick $Si_3N_4$ layer is deposited as a dielectric layer to form the pixel capacitor. A 100 nm thick aluminum gate electrode is deposited by sputtering and patterned using the gate-mask process. The SiNx gate dielectric layer is deposited via plasma-enhanced chemical vapor deposition. Next, the a-Si:H layer is patterned as an active layer using the channel-mask process. Molybdenum source and drain electrodes are deposited by sputtering and then patterned using the electrode-mask process to form double-channel, bottom-gate a-Si TFTs. A 2 μm thick polymer layer is coated as a passivation layer, followed by the formation of holes using the via-mask process. In the final step, the pixel electrode is defined by sputtering ITO and patterning it with the pixel-mask process.

### Time-domain THz spectroscopy
We measured the intensity and phase of THz light using a time-domain THz spectrometer, TeraFlash by Toptica Photonics, capable of acquiring spectra at a rate of 16 spectra/s. This spectrometer facilitates the THz reflection characterization of the device across a spectral range of up to 4 THz and with over a 90 dB dynamic range. Two InGaAs fiber-coupled antennas, along with the THz optical path, are housed within a 4-mirror reflection head that is continuously purged with dry nitrogen. This setup eliminates water absorption lines from the THz spectra during reflectivity measurements. The device is mounted on two perpendicular X and Y motorized stages (Thorlabs linear translational stages), allowing movement in the plane perpendicular to the THz beam, while keeping the beam focused on the modulator surface. A custom LabView code simultaneously controls the THz spectrometer, the stages and the THz modulator. Raster scanning is performed in 0.5 mm steps. The system moves the modulator and acquires the reflection spectrum, while the voltage on modulator array alternates between $V_{DH}$ and $V_{DL}$. We obtain the THz amplitude and phase spectrum using an FFT algorithm with appropriate zero padding to achieve a spectral resolution of 1 GHz.

### Imaging with THz camera
We assessed the modulator's performance and its capability to generate complex patterns using a Terasense camera (based on plasmon oscillations on GaAs high mobility heterostructure) with a $64 \times 64$

array with a 1.5 mm pixel pitch. The modulator was illuminated with a collimated 100 GHz beam produced by an IMPATT diode, Terasense, 80 mW output power. To minimize light scattering, the modulator was placed directly over the THz camera array.

## Single pixel THz camera

We developed a THz imaging system utilizing a continuous-wave 100 GHz beam from an IMPATT diode, Terasense, 80 mW output power. This beam, collimated by an 8 in. parabolic mirror or Teflon lens, passes through both the modulator and the object. It is then focused by another 8 in. parabolic mirror or Teflon lens into a pyroelectric detector, Gentec-EO model. A Raspberry Pi 3 Model B served as the electronic controller, transmitting Hadamard patterns to the modulator. Signal reading from the pyroelectric detector and synchronization with the Raspberry Pi and THz spectrometer were managed by a LabView-based FPGA and microprocessor board, myRIO-1900 from National Instruments.

## Electromagnetic simulations

Using COMSOL-multiphysics for electromagnetic simulation, we modeled the charge density on graphene by solving modified PNP equations. These simulations demonstrated that charge density on continuous graphene can be modulated at a single pixel level. The electrolyte layer's thickness and pixel coverage are crucial to minimize crosstalk effects between pixels on the graphene layer.

## Data availability

All relevant data discussed in the main text is available at Zenodo[37].

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

## Acknowledgements

This research is supported by the Defense Science and Technology Laboratory (DSTLX–1000135951) and UKRI EP/X027643/1 (ERC PoC grant). We thank Prof. Askin Kocabas for the help of the electromagnetic simulations.

## Author contributions

Y.M., M.S.E., G.B., and C.K. designed and fabricated the devices. Y.M. and M.S.E. built the setups and performed the experiments. G.B. developed the code for the TFT array and performed the electrical characterization of the TFTs. P.S. helped with the experiments and sample preparation. Y.M., M.S.E., and C.K. analyzed the data and wrote the manuscript with feedback from the other authors. All authors discussed the results and contributed to the scientific interpretation as well as to the writing of the manuscript.

## Competing interests

Y.M., M.S.E., G.B., and C.K. are involved in a patent application by the University of Manchester (PCT/GB2022/051954). Other authors do not have any competing interests. There are no more competing interests.
