## [Transparent Peer Review file · Nature Communications]

Very-Large-Scale Reconfigurable Intelligent Surfaces for Dynamic Control of Terahertz and Millimetre Waves

Corresponding Author: Professor Coskun Kocabas

This manuscript has been previously reviewed at another journal that is not operating a transparent peer review scheme. The manuscript was considered suitable for publication without further review at Nature Communications.

Version 0:

Reviewer comments:

Reviewer #1

(Remarks to the Author)

The authors present an electronically programmable terahertz spatial light modulator (SLM) with 640×480 pixels. Subsequently, single-pixel mm-wave imaging of metallic objects using the SLM was performed. To my knowledge, this is the electronic terahertz SLM with the largest number of pixels reported. In my opinion, this manuscript could be considered for publication in *Nature Communications* after addressing the following revisions:

1. The biggest challenge in terahertz single-pixel imaging lies in the fabrication of the SLM. For electronic SLMs, fabricating large-scale electrode arrays remains very challenging. This manuscript demonstrates a graphene-integrated TFT technology to fabricate a large-scale device. However, the fabrication details are scarcely discussed in both the main text and the supplementary information. Given that this is a key innovation of the paper, a detailed explanation is essential.
2. As we know, terahertz light has difficulty penetrating conductive layers. What is the transmission loss of the SLM compared to the scenario without the SLM?
3. For graphene devices, especially those gated with electrolytes, the Dirac point should be observed. This information should be included in the main text or the supplementary material.
4. What is the imaging speed of the transmission single-pixel imaging in Fig. 4? Given that the charging time of a pixel is at the millisecond level, as shown in Fig. 1f, this information is important.
5. It appears that the SLM has better modulation in the reflection geometry, as shown in Fig. 3. Why not perform single-pixel imaging in the reflection configuration using a terahertz time-domain spectroscopy system?
6. It would be beneficial to perform single-pixel imaging at more than one frequency in the transmission geometry, as shown in Fig. 4.

Reviewer #2

(Remarks to the Author)

This manuscript presents a 300 k-pixel terahertz spatial light modulator implemented by capacitively coupling the an array of individually-addressed thin-film transistors to a uniform graphene sheet, allowing for highly localized modulation of sheet conductance. The authors deploy this device to demonstrate a series of transmission and reflection masks, as well as coded-aperture imaging. The structure in question is novel and potentially, the results will be of interest to the research community, and the report itself is timely. However, there are several issues that must be addressed in order for this manuscript to merit publication as a journal article.

1. Firstly, and this is a relatively minor point, but paper abstracts are generally not supposed to include citations. References 1 to 7 have been cited in the abstract of this paper, and these citations should be removed.
2. Please provide an explanation for the choice of 100 GHz as operation frequency for the transmissive spatial-light modulator demonstrations presented in Fig. 2(b) and Fig. 4. Furthermore, given this particular choice of frequency for these demonstrations, it is necessary to include the voltage-dependence of transmission at 100 GHz in Fig. 2(d).

3. In the conclusion, the authors state that “Our results suggest that these surfaces can be used as a phased array or spatial light modulator to dynamically steer THz light or to create complex beams which is a critical capability required for next generation communication systems.” This claim of phased-array operation is not an accurate statement given that Fig. (b) shows that the desired phase shift is accompanied by severe losses beyond 20 dB, which renders it highly unsuitable to act as a phase mask. Indeed, the authors have even used the term “perfect absorption” to describe these losses. Furthermore, although the phase switching range is indeed large, the fact that this takes the form of a “step-like jump” essentially renders intermediate phase levels inaccessible. This leads to very coarse quantization of the intended phase mask, further casting operation as a phased array further into doubt. This inaccurate claim must therefore be removed, and all mention of phased-control operation must be heavily revised in order to be a more accurate representation of the capabilities of this spatial light modulator.

The reported device is a perfectly fine magnitude-based spatial light modulator. There is no need to overextend claims of its capabilities beyond that.

4. The abstract states that the device is “mega-scale,” but there is no precise definition of what this means. It would be more accurate to name it “kilo-scale,” given that there are $640 \times 480 = 300k$ -pixels. This statement must be revised.

5. Are the results in Fig. 2(b) with single, double, or triple? What about Fig. 2d,e, Fig. 3b-f, or Fig. 4b-d? Only Fig. 2(c) specifies the number of graphene layers used, and this must be rectified.

Reviewer #3

(Remarks to the Author)

I have read “Very-Large-Scale Reconfigurable Intelligent Surfaces for Dynamic Control of Terahertz and Millimetre Waves” by Professor Kocabas and colleagues. This manuscript presents an experimental study of a large format terahertz (THz) spatial light modulator operating in both transmission and reflection-based modes that also explores the use for wavefront control, including applications towards single pixel imaging. To accomplish this, they have taken the core graphene device previously developed nearly a decade ago and integrated it with a commercial thin-film transistor (TFT) technology to enable spatial and temporal control. The main accomplishment of the work is the integration of the two technologies, which is clearly demonstrated within the work. Overall, the author’s work presents relatively modest operational performance in terms of spectral and temporal characteristics in comparison to other approaches that have been demonstrated (most notably that of Venkatesh [1]), and thus, I am unable to find sufficient novelty that warrants a recommendation for publication in Nature Communications.

I have included additional questions/comments below to help clarify and strengthen the work:

The proposed device behaves much more as a spatial light amplitude modulator (SLM) than as what the authors refer to as Reconfigurable Intelligent Surface (RIS) due to the high insertion loss on resonance. Perhaps reconsider referencing this simply as an SLM vs RIS.

Is the same device being used for transmission and reflectance? If not, what is different between them and what are the design objectives (e.g., maintaining low insertion loss, complete phase coverage, amplitude dynamic range)?

There are missing details related to the transmission mode of operation. Typically, the electrodes for SLMs prevent low insertion loss, and there aren’t any details with regards to what the absolute transmission is, how it performs both spectrally, polarization dependence, and field of view behavior. I recommend including these details in the main text. The plots in Figure 2 (c-e) were confusing, is this absolute change in transmittance, modulation depth?

The demonstrator shown in Figure 2 is oddly placed. I would consider removing it to later in the text or placing within the supplementary.

For temporal modulation, a more direct measurement of speed would be desirable whereby you show peak to peak maximum modulation as a function of modulation frequency.

The grey scale control via trimming the electrical pulse is interesting, and a better description in the text to understand the different contributions (graphene properties, thickness of the polymer, and other parasitics in the control electronics) and how they influence the ultimate response times. There are details in the supplementary that can perhaps be summarized in the main text, along with complimentary figures.

The results presented reference the use many different frequency points which is confusing to track. I would recommend using the same set of frequencies to avoid confusion.

In Figures 3b and c the amplitude and phase plots would benefit from a legend with regards to the voltages (consider reducing the number of curves as well). The overall insertion loss across the wavelengths of interest are high, where is this coming from, the graphene or something else? Is there a path for this to be reduced? The phase plot in 3c is difficult to understand what the phase coverage is, please correct the branch error to more clearly map the phase coverage.

The description and discussion regarding imaging is not sufficient. More details are needed regarding the setup and implementation. The wavelength being 3mm is quite long relative to the pixels the claim of imaging features that are subwavelength suggests the object may be acting as a shadow mask vs something more in line with a typical conjugate image which is spatially sampled.

References

1. Venkatesh, S., Lu, X., Saeidi, H. et al. A high-speed programmable and scalable terahertz holographic metasurface based on tiled CMOS chips. *Nat Electron* 3, 785–793 (2020). <https://doi.org/10.1038/s41928-020-00497-2>

Version 1:

Reviewer comments:

Reviewer #1

(Remarks to the Author)

The authors have addressed all the previously listed questions. No further comments are necessary. I recommend the acceptance of the manuscript.

Reviewer #2

(Remarks to the Author)

The authors have addressed all of my comments to my satisfaction, and I am willing to endorse this manuscript for publication in *Nature Communications*.

Reviewer #3

(Remarks to the Author)

I have reviewed the revisions by the authors, and appreciate the additional details that they have included within both the response and revised manuscript. The manuscript has been greatly improved and the significance of their finding is much more evident. That being said, I would still recommend removal of the use of 'Reconfigurable Intelligent Surfaces' given the demonstration being limited to its use as an amplitude based spatial light modulator (SLM). The phase coverage demonstrated across the wavelengths is also quite limited, which further limits the strength of that claim. They also did not exactly what the source of losses are coming from in the transmission mode, and if it is in fact the electrode array or the active layer itself. The ability to create large scale electrodes that are highly transparent is extremely novel, and if the authors were able show evidence that would eventually be possible within their platform, there would be immense benefit to the community.

All modifications have been highlighted on the main text.

REVIEWER COMMENTS

Reviewer #1 (Remarks to the Author):

*The authors present an electronically programmable terahertz spatial light modulator (SLM) with 640×480 pixels. Subsequently, single-pixel mm-wave imaging of metallic objects using the SLM was performed. To my knowledge, this is the electronic terahertz SLM with the largest number of pixels reported. In my opinion, this manuscript could be considered for publication in *Nature Communications* after addressing the following revisions:*

Our response: We thank the reviewer for the favourable comments. We have revised the manuscript to address the points raised by the reviewers.

1. The biggest challenge in terahertz single-pixel imaging lies in the fabrication of the SLM. For electronic SLMs, fabricating large-scale electrode arrays remains very challenging. This manuscript demonstrates a graphene-integrated TFT technology to fabricate a large-scale device. However, the fabrication details are scarcely discussed in both the main text and the supplementary information. Given that this is a key innovation of the paper, a detailed explanation is essential.

Our response: We have provided more details of the fabrication processes.

Modification to the manuscript:

The backplane electronics are fabricated on a 500 μm thick glass substrate using a conventional a-Si TFT process. The design consists of four electrode layers (back electrode, gate electrode, source-drain electrode, and pixel electrode), two dielectric layers (capacitor dielectric and gate dielectric), and one passivation layer. The first back electrode layer is formed by sputtering and patterning a 100 nm thick aluminium layer. A 250 nm thick SiNx layer is deposited as a dielectric layer to form the pixel capacitor. A 100 nm thick aluminium gate electrode is deposited by sputtering and patterned using the gate-mask process. The SiNx gate dielectric layer is deposited via plasma-enhanced chemical vapor deposition (PECVD). Next, the a-Si layer is deposited and patterned as an active layer using the channel-mask process. Molybdenum source and drain electrodes and top capacitor electrode are deposited by sputtering and then patterned using the electrode-mask process to form double-channel,

bottom-gate a-Si TFTs. A 2 μm thick polymer layer is coated as a passivation layer, followed by the formation of holes using the via-mask process. In the final step, the pixel electrode is defined by sputtering indium tin oxide (ITO) and patterning it with the pixel-mask process.

2. As we know, terahertz light has difficulty penetrating conductive layers. What is the transmission loss of the SLM compared to the scenario without the SLM?

Our response and modification to the manuscript:

Yes, the THz transmission of the modulator is limited by the conductive layers, capacitor electrode and the top ITO layer. To increase transmission we have used smaller capacitor and thin ITO layer. The following figure shows the transmission spectrum of the device for vertical and horizontal polarization. The modulator shows around 30% transmission at 100 GHz for horizontal polarization and around 20% for vertical polarization. The transmission decreases significant at higher frequencies. At 1 THz, both polarizations show less 5% transmission. The transmission measurements presented in the paper were recorded using 100 GHz source with horizontal polarization.

We have added a new figure and discussion in the supporting materials.

Figure S5: Transmission spectrum of the modulator array. The inset shows the picture of the TFT array to provide reference for the direction of polarization. The transmission measurements presented in the paper were recorded using 100 GHz source with horizontal polarization

3. For graphene devices, especially those gated with electrolytes, the Dirac point should be observed. This information should be included in the main text or the supplementary material.

Our response and modification to the manuscript: Indeed, the electrostatic gating of graphene shows a Dirac point at the lowest charge density. We have added a figure showing the variation of the reflection at non-resonant frequency which clearly shows the charge neutrality point at 0.1 V. We have also observed that the Dirac point could shift to higher voltages after operation due to some electrochemical effects.

Figure S8: The variation of the reflection at 1THz as a function of pixel voltage. It should be noted this this measurement obtain in the static mode by grounding the electrodes and changing the voltage applied to the graphene. Due to the complexity of the control electronics and the active-matrix circuit, the drain voltage does not appear directly on the pixel due to charging time.

4. What is the imaging speed of the transmission single-pixel imaging in Fig. 4? Given that the charging time of a pixel is at the millisecond level, as shown in Fig. 1f, this information is important.

Our response: The speed of single-pixel imaging depends on various factors, including the refresh rate of the TFT array, the communication speed of the driver chip, and the inherent speed of the graphene modulators. The timings for these processes are illustrated in the following figure. The slowest step is the data transmission to the driver chip, which takes approximately 100 ms. Once the pattern is updated on the modulator, the scan time of the TFT array ranges from 5–20 ms, while the pixel charging time is around 1–2 ms. The pattern on the modulator can be refreshed at a rate of 5 image per second.

For the single-pixel images presented in Figure 4, we used 1024 masks, requiring approximately 5 minutes to complete. The primary limiting factor for imaging speed is the processing capacity of the display chip. Enhancing the imaging speed is possible by using more powerful display drivers.

We have included the corresponding figure in the supporting materials.

Figure 10: Time trace of the transmitted signal during the pattern update. The speed of single-pixel imaging depends on various factors, including the refresh rate of the TFT array, the communication speed of the driver chip, and the inherent speed of the graphene

modulators. The following figure illustrates these timings. The slowest step is data transmission to the driver chip, which takes around 100 ms. After updating the pattern on the modulator, the scan time of the TFT array is between 5–20 ms. The pixel charging time is approximately 1–2 ms. The pattern on the modulator can be refreshed at a rate of frames per second. For the single-pixel images shown in Figure 4, we used 1024 masks, which took around 5 minutes to complete. The main limiting factor for imaging is the processing capacity of the display driver chip. The imaging speed can be increased by using more powerful display drivers.

5. It appears that the SLM has better modulation in the reflection geometry, as shown in Fig. 3. Why not perform single-pixel imaging in the reflection configuration using a terahertz time-domain spectroscopy system?

Our response: We agree with the referee that the reflection performance of the device is superior to its transmission modulation capabilities. However, we were unable to achieve single-pixel imaging in the reflection mode due to technical limitations of the time-domain system. Specifically, the size of the optical components and the power levels has limited us. In our current time-domain setup, unfortunately, the available power levels were insufficient to overcome the losses associated with the reflection configuration. Moreover, the dimensions of the optical components in the time-domain system were not enough to cover the large area modulator needed for single-pixel imaging in the reflection mode.

6. It would be beneficial to perform single-pixel imaging at more than one frequency in the transmission geometry, as shown in Fig. 4.

Our response: We agree with the referee that spectroscopic imaging would provide greater insight into the imaged object. However, due to the limitations of our experimental setup, we were unable to perform imaging in reflection mode. In the transmission mode, we are constrained to using the only available high power source a 100 GHz source and available detector pair, which restricts our capability to a single frequency.

Reviewer #2 (Remarks to the Author):

This manuscript presents a 300 k-pixel terahertz spatial light modulator implemented by capacitively coupling the an array of individually-addressed thin-film transistors to a uniform graphene sheet, allowing for highly localized modulation of sheet conductance. The authors deploy this device to demonstrate a series of transmission and reflection masks, as well as coded-aperture imaging. The structure in question is novel and potentially, the results will be of interest to the research community, and the report itself is timely. However, there are several issues that must be addressed in order for this manuscript to merit publication as a journal article.

1. Firstly, and this is a relatively minor point, but paper abstracts are generally not supposed to include citations. References 1 to 7 have been cited in the abstract of this paper, and these citations should be removed.

Our response: We moved the references from abstract to first paragraph of the main text.

2. Please provide an explanation for the choice of 100 GHz as operation frequency for the transmissive spatial-light modulator demonstrations presented in Fig. 2(b) and Fig. 4. Furthermore, given this particular choice of frequency for these demonstrations, it is necessary to include the voltage-dependence of transmission at 100 GHz in Fig. 2(d).

Our response: There are couple of technical reasons behind the choice of 100 GHz as operation frequency. (1) sensitivity of the THz camera. We used TeraSense camera which is based on resonance plasmon effects on 2D electron gas. The sensitivity of the camera is optimised for very narrow spectral range around 100GHz. (2) The frequency range of the available detector and W10 waveguides. These factors restrict us to choose demonstrations at 100GHz.

We have provided voltage dependent transmission at 100GHz in supporting materials.

Figure S6: Voltage dependent transmission of double layer graphene modulator shown in Figure 2. This curve is obtained at static mode, by applying external voltage to the graphene layers as the pixel electrodes are grounded. It should be noted that the effective voltage during the operation of the modulator depends on many factors such as refresh rate of the modulator, drain voltage and pulse duration. Therefore, drain voltage and external voltage are not the same quantity.

3. In the conclusion, the authors state that "Our results suggest that these surfaces can be used as a phased array or spatial light modulator to dynamically steer THz light or to create complex beams which is a critical capability required for next generation communication systems." This claim of phased-array operation is not an accurate statement given that Fig. (b) shows that the desired phase shift is accompanied by severe losses beyond 20 dB, which renders it highly unsuitable to act as a phase mask. Indeed, the authors have even used the term "perfect absorption" to describe these losses. Furthermore, although the phase switching range is indeed large, the fact that this takes the form of a "step-like jump" essentially renders intermediate phase levels inaccessible. This leads to very coarse quantization of the intended phase mask, further casting operation as a phased array further into doubt. This inaccurate claim must therefore be removed, and all mention of phased-control operation must be heavily revised in order to be a more accurate representation of the capabilities of this spatial light modulator.

The reported device is a perfectly fine magnitude-based spatial light modulator. There is no need to overextend claims of its capabilities beyond that.

Our response: We agree with the referee that our devices are not operating as a conventional phased array. Therefore, we have removed the term "phased array" and refer our devices as spatial light modulator. We have revised the discussion on the phase modulation with more emphasis on the link between phase and intensity modulation.

The phase jump obtained by the reflection singularity is a new type of phase modulation which has not been explored widely. This phase modulation is due to a topological switching of the complex Fresnel reflection. In our recent works (ref 33), we have reported this effect. We believe that a short discussion on this is beneficial to understand the reflection modulation.

4. The abstract states that the device is "mega-scale," but there is no precise definition of what this means. It would be more accurate to name it "kilo-scale," given that there are $640 \times 480 = 300k$ -pixels. This statement must be revised.

Our response:

We agree with the referee that the term "mega-scale" may be misleading. We have removed it from the abstract and added a new figure to illustrate the potential of our method to fabricate megapixel modulator arrays. The scalability of our approach is primarily constrained by the capabilities of modern display technology, which enables the production of very large-scale, dense arrays of TFTs on both rigid and flexible substrates.

In this work, we characterized a 0.3-megapixel TFT device. However, our method is capable of scaling the device size to multiple megapixels. To demonstrate this scalability, we have included a new figure in the Supporting Materials. This figure highlights the feasibility of extending our approach to create larger arrays, emphasizing the practical potential of our method for future applications.

Figure S3: Megapixel THz modulator arrays: Photograph of 2.5 megapixel (1800x1400) of TFT arrays on glass and flexible Kapton substrates. The pixel size is 0.11x0.11 mm. The scale bar shows 0.3 mm. The inset shows the individual pixels and complex wiring. The scalability of our approach is primarily constrained by the capabilities of modern display technology, which enables the production of very large-scale, dense arrays of TFTs on both rigid and flexible substrates. In the manuscript, we only characterized a 0.3-megapixel TFT device. However, our method is capable of scaling the device size to multiple megapixels with suitable back plane electronics.

5. Are the results in Fig. 2(b) with single, double, or triple? What about Fig. 2d,e, Fig. 3b-f, or Fig. 4b-d? Only Fig. 2(c) specifies the number of graphene layers used, and this must be rectified.

Our response: All the results presented in the paper is based on modulators fabricated using double layer graphene which shows the highest modulation with lowest insertion loss. We have clarified this point in the manuscript.

Reviewer #3 (Remarks to the Author):

I have read "Very-Large-Scale Reconfigurable Intelligent Surfaces for Dynamic Control of Terahertz and Millimetre Waves" by Professor Kocabas and colleagues. This manuscript presents an experimental study of a large format terahertz (THz) spatial light modulator operating in both transmission and reflection-based modes that also explores the use for wavefront control, including applications towards single pixel imaging. To accomplish this, they have taken the core graphene device previously developed nearly a decade ago and integrated it with a commercial thin-film transistor (TFT) technology to enable spatial and temporal control. The main accomplishment of the work is the integration of the two technologies, which is clearly demonstrated within the work. Overall, the author's work presents relatively modest operational performance in terms of spectral and temporal characteristics in comparison to other approaches that have been demonstrated (most notably that of Venkatesh [1]), and thus, I am unable to find sufficient novelty that warrants a recommendation for publication in Nature Communications.

I have included additional questions/comments below to help clarify and strengthen the work:

The proposed device behaves much more as a spatial light amplitude modulator (SLM) than as what the offers refer to as Reconfigurable Intelligent Surface (RIS) due to the high insertion loss on resonance. Perhaps reconsider referencing this simply as an SLM vs RIS.

Our response: We agree with the referee that our device can be considered both as a spatial light modulator (SLM) and as a Reconfigurable Intelligent Surface (RIS). The ability to fabricate large-scale TFT arrays makes our approach particularly well-suited for RIS applications, which require extensive spatial modulation.

In the paper we acknowledge that the insertion loss of the device, particularly on resonance, limits its applicability in certain scenarios. Reconfigurable Intelligent Surface for microwave frequencies in the literature has very similar performance in terms of reflection and phase modulation and insertion loss around the resonance. However, this limitation can potentially be addressed by integrating the device with frequency-selective surfaces, or meta surfaces which could significantly reduce the loss. This enhancement would expand the range of practical applications for the device in both SLM and RIS contexts.

We agree with the referee that the recommended paper demonstrates impressive performance at 300 GHz with <2 mm active device, However, scalability of CMOS based approach remains a challenge for large-area implementations, particularly in the context

of RIS applications, where uniform performance and manufacturability over a wide area are critical.

We believe that our approach provides a valuable complementary solution, particularly for applications requiring large-scale modulation where other methods may face limitations. By integrating modern TFT technology with active THz materials, the scalability challenges could potentially be addressed in future research. We are confident that this work contributes significantly to the broader field and opens up opportunities for THz applications.

Is the same device being used for transmission and reflectance? If not, what is different between them and what are the design objectives (e.g., maintaining low insertion loss, complete phase coverage, amplitude dynamic range)?

Our response: Yes, the same device is used for both transmission and reflection modulation. While the fundamental design remains the same, the performance in each mode could be significantly enhanced by optimizing the pixel structure and selecting substrates tailored to specific applications. However, the high cost of TFT design and fabrication currently poses a limitation to implementing such optimizations at a university lab.

There are missing details related to the transmission mode of operation. Typically, the electrodes for SLM's prevent low insertion loss, and there aren't any details with regards to what the absolute transmission is, how it performs both spectrally, polarization dependence, and field of view behavior. I recommend including these details in the main text.

The plots in Figure 2 (c-e) were confusing, is this absolute change in transmittance, modulation depth?

Our response: We have provided further details of the transmission spectrum of the devices in supporting materials. We have added a new figure and discussion in the supporting materials. The modulation presented in Figure 2 is normalised modulation based on the transmitted intensity. We have clarify this in the paper.

Figure S5: Transmission spectrum of the modulator array. The inset shows the picture of the TFT array to provide reference for the direction of polarization. The transmission measurements presented in the paper were recorded using 100 GHz source with horizontal polarization. The THz transmission could be enhanced with special pixel design to enable less surface covered by capacitance and top ITO layer.

The demonstrator shown in Figure 2 is oddly placed. I would consider removing it to later in the text or placing within the supplementary.

Our response: We aim to explain the reflection modulation later in the manuscript due to the complex interference effects involved. For this reason, we have positioned Figure 2 before the reflection characterization section.

For temporal modulation, a more direct measurement of speed would be desirable whereby you show peak to peak maximum modulation as a function of modulation frequency.

The grey scale control via trimming the electrical pulse is interesting, and a better description in the text to understand the different contributions (graphene properties, thickness of the polymer, and other parasitics in the control electronics) and how they influence the ultimate response times. There are details in the supplementary that can perhaps be summarized in the main text, along with complimentary figures.

Our response: Due to the complexity of the control circuit and 300k pixels, it is not possible to measure the peak to peak maximum modulation as a function of modulation frequency. However we have measured variation of the modulation as a function of the number of pulse applied to the pixels.

Figure S7: The variation of the transmission modulation as a function of the number of applied pulses to the pixel. Here the pulse width is 30 μ sec. The increasing number of pulses increases the modulation depth, however it reduces the total time to refresh the pattern on the modulator.

We have also added a new graph showing timing of the data transmission, row scan and charging of the pixels.

Figure 5: Time trace of the transmitted signal during the pattern update. The speed of single-pixel imaging depends on various factors, including the refresh rate of the TFT array, the communication speed of the driver chip, and the inherent speed of the graphene modulators. The following figure illustrates these timings. The slowest step is data transmission to the driver chip, which takes around 100 ms. After updating the pattern on the modulator, the scan time of the TFT array is between 5–20 ms. The pixel charging time is approximately 1–2 ms. The pattern on the modulator can be refreshed at a rate of frames per second. For the single-pixel images shown in Figure 4, we used 1024 masks, which took around 5 minutes to complete. The main limiting factor for imaging is the processing capacity of the display chip. The imaging speed can be increased by using more powerful display drivers.

The results presented reference the use many different frequency points which is confusing to track. I would recommend using the same set of frequencies to avoid confusion.

Our response: We have used only two frequency, 100 GHz for transmission characterisation and 1.2-2.2 THz range for reflection characterization. We have clarified these points.

In Figures 3b and c the amplitude and phase plots would benefit from a legend with regards to the voltages (consider reducing the number of curves as well). The overall insertion loss across the wavelengths of interest are high, where is this coming from, the graphene or something else? Is there a path for this to be reduced? The phase plot in 3c is difficult to understand what the phase coverage is, please correct the branch error to more clearly map the phase coverage.

Our response: We have added a legend to the graph for better clarity. The insertion loss of the device in reflection mode arises due to the resonance effect. This resonance is caused by the interference of multiple reflected beams originating from the top layer, the graphene layer, and the bottom pixel electrode. The interplay of these reflections leads to the observed insertion loss mainly due to the enhanced absorption of the graphene layer.

Additionally, the phase jump observed in the measurements is not due to a branch error but has a topological origin. This large phase modulation due to topological switching presents an exciting opportunity to achieve control of reflection across a broad spectrum (see ref 33). Future optimization of the device structure is needed, including the design of the reflective layers and resonance tuning, could further enhance its performance and expand its application potential.

The description and discussion regarding imaging is not sufficient. More details are needed regarding the setup and implementation. The wavelength being 3mm is quite long relative to the pixels the claim of imaging features that are subwavelength suggests the object may be acting as a shadow mask vs something more in line with a typical conjugate image which is spatially sampled.

Our response: We appreciate the referee's feedback and have expanded the description of the imaging setup and implementation in the revised manuscript. The claim of subwavelength imaging refers to the spatial resolution achieved through computational reconstruction where the object is placed on the modulator. At a wavelength of 3 mm, the object acts as a shadow mask, and the recorded intensity with the Hadamard patterns are used to computationally reconstruct the spatial features, as detailed in the updated text and supporting

References

1. Venkatesh, S., Lu, X., Saeidi, H. et al. A high-speed programmable and scalable terahertz holographic metasurface based on tiled CMOS chips. Nat Electron 3, 785–793 (2020). <https://doi.org/10.1038/s41928-020-00497-2> [doi.org]

Our response: We have added this reference as Ref 15.

REVIEWERS' COMMENTS

Reviewer #1 (Remarks to the Author):

The authors have addressed all the previously listed questions. No further comments are necessary. I recommend the acceptance of the manuscript.

Our response: We thank the favourable comments and recommendation for publications.

Reviewer #2 (Remarks to the Author):

The authors have addressed all of my comments to my satisfaction, and I am willing to endorse this manuscript for publication in Nature Communications.

Our response: We thank the favourable comments and endorsing for publications.

Reviewer #3 (Remarks to the Author):

I have reviewed the revisions by the authors, and appreciate the additional details that they have included within both the response and revised manuscript. The manuscript has been greatly improved and the significance of their finding is much more evident.

Our response: We thank the favourable comments.

That being said, I would still recommend removal of the use of 'Reconfigurable Intelligent Surfaces' given the demonstration being limited to its use as an amplitude based spatial light modulator (SLM). The phase coverage demonstrated across the wavelengths is also quite limited, which further limits the strength of that claim. They also did not exactly what the source of losses are coming from in the transmission mode, and if it is in fact the electrode array or the active layer itself. The ability to create large scale electrodes that are highly transparent is extremely novel, and if the authors were able show evidence that would eventually be possible within their platform, there would be immense benefit to the community.

Our response: We agree with the referee that our manuscript does not address all the challenges of RIS. We also thank the referee for the suggestion to change the title, however we believe that 'Reconfigurable Intelligent Surfaces' is more suitable for our applications. Spatial light modulators are specifically used for small scale beam forming. In this work we demonstrate a new approach which can be scaled to very large areas using modern display technology. RIS applications require very large area devices. This makes our approach particularly important for RIS. Therefore we want to keep the 'Reconfigurable Intelligent Surfaces' which will make a bigger impact in the community.